# Study on Flowering Dynamics and Pollination Habits of *Monochasma savatieri* Under Artificial Cultivation Conditions

**DOI:** 10.3390/plants14050715

**Published:** 2025-02-26

**Authors:** Haoqian Zhang, Qiuling Wang, Jianhe Wei

**Affiliations:** State Key Laboratory of Bioactive Substance and Function of Natural Medicines & National Engineering Laboratory for Breeding of Endangered Medicinal Materials, Institute of Medicinal Plant Development, Chinese Academy of Medical Sciences & Peking Union Medical College, Beijing 100193, China; zhanghq00@foxmail.com (H.Z.); qlwang@impladac.cn (Q.W.)

**Keywords:** *Monochasma savatieri*, flowering dynamics, pollination habit

## Abstract

The research aimed to investigate the reproductive biology of *M. Savatieri* (*Monochasma savatieri* Franch. ex Maxim.), focusing on its floral structure, developmental stages, flowering dynamics, pollen dispersal characteristics, stigma development, and longevity, as well as self-crossing affinity in both indoor and outdoor cultivation conditions. The study divided the flower development process into four distinct stages: the bud stage (1–3 days), the present bud stage (4–8 days), the pollination stage (9–14 days), and the fruiting stage (15–35 days). Pollen dispersal begins one hour after flowering during the pollination stage and ends within five hours under both indoor and outdoor sunny conditions. Pollen remains highly viable for up to 60 days. Stigma receptivity reaches its maximum at 94.44% on the first day of flowering, decreases to 68.2% by the fourth day, and further declines to 16.7% on the fifth day. The plant, with an outcrossing index (OCI) of 3, shows a 28.80% outcrossing rate in indoor environments and 9.57% in outdoor conditions. It primarily relies on heterogamous pollination while being capable of both self-pollination and artificial pollination. The results provide significant insights into the breeding and production of *M. savatieri*.

## 1. Introduction

*M. Savatieri (Monochasma savatieri* Franch. ex Maxim.) is a semi-parasitic perennial herb belonging to the Scrophulariaceae family. It is distributed across multiple provinces in China, including Zhejiang, Jiangxi, Fujian, Anhui, Jiangsu, Hunan, and Guangdong, commonly found along forest roadsides and in thickets. This species is also reported in Japan.

*M. savatieri*, a traditional Chinese medicinal herb, is used holistically to treat conditions such as respiratory infections, inflammatory disorders, and gynecological ailments [1]. Its key bioactive components—flavonoids, alkaloids, saponins, and polysaccharides [2,3,4]—contribute to broad pharmacological activities, including antimicrobial, antioxidant, and anti-inflammatory effects [5,6,7,8,9,10,11,12,13,14]. Recent studies emphasize its anticancer potential [15].

In recent years, the decline in wild resources of *M. savatieri* and the challenges in procuring raw materials for the pharmaceutical industry have necessitated a shift toward its artificial cultivation. The author’s research team has played a key role in transitioning *M. savatieri* from a wild to a domesticated species, successfully addressing challenges related to seedling development, transplantation, and large-scale cultivation. A base for the artificial cultivation of *M. savatieri* and a germplasm resource nursery has also been established.

However, production challenges remain, including significant individual variability, inconsistent yield and quality, and notable differences in disease resistance. Understanding the reproductive biology of *M. savatieri* is essential for variety selection and the development of effective cultivation techniques. Unveiling the biological aspects of flowers, such as floral characteristics, the lifespan of flowering organs, and the extent of self-crossing affinity, is critical for breeding improvement. In the study of the Scrophulariaceae family, the reproductive system of *Scrophularia ningpoensis* Hemsl is partially self-compatible, requiring pollinators for outcrossing with a natural outcrossing rate of 72.15%, classifying it as a typical outcrossing plant [16]. Previous research has explored the flowering traits of *M. savatieri* in natural habitats [17], but limited studies have examined its flowering dynamics and pollination characteristics under cultivated conditions. Additionally, research addressing the effects of different artificial cultivation environments is scarce.

This study conducted an in-depth investigation into the flowering dynamics, pollination characteristics, stigma development and longevity, and self-incompatibility of cultivated *M. savatieri* grown from seed. The findings hold significant value for the selection and breeding of *M. savatieri*.

## 2. Results

### 2.1. Observations on the Appearance and Flowering Dynamics of M. savatieri Flowers

#### 2.1.1. Appearance and Morphology of *M. savatieri* Flowers

In the Flora of China [18], detailed records have been made concerning the morphology and size of each part of *M. savatieri* flowers, while the color of the floral organs has been less documented. In this experiment, the size and color of each floral part were recorded by observing traits in indoor cultivated specimens.

*M. savatieri* produces terminal racemes with branches that are glandular-hairy and possess wooly hairs near the flowers. The flowers are sparsely distributed, typically solitary in the leaf axils, and are borne on short pedicels measuring 0.15 to 0.55 cm in length. Each flower has two leafy bracteoles measuring 0.69 to 1.21 cm in length and 0.15 to 0.25 cm in width, located at the base of the calyx tube. The calyx tube is membranous and wooly in texture, occasionally interspersed with glandular hairs. It measures 0.90 to 1.95 cm in length and displays nine prominent ribs, with four extending into the calyx teeth (Figure 1e). The calyx teeth, numbering four, are herbaceous, linear to linear-lanceolate with an acuminate apex; they are about equal in length to the calyx tube, measuring 0.10 to 0.30 cm in width (Figure 1b).

The corolla primarily exhibits a white or light pink coloration, with darker pink sections. The upper lip is two-merous, while the lower lip is three-merous and lighter in color (Figure 1c). The exterior of the floral tube has a deeper hue, and the corolla progressively darkens as the flower matures and blooms.

The basal section of the floral tube features a yellowish band extending to the opening of the lower lip, forming a two-pointed yellow region highlighted by a few orange spots (Figure 1d). In flowers with darker pink corollas, the band and spots appear orange-red. The exterior of the corolla exhibits subtle pilosity. The floral tube is slender, expanding near the throat, and measures about twice the length of the calyx, ranging from 0.90 to 1.95 cm in length, 1.15 to 1.92 cm in height, and 1.20 to 2.20 cm in width. The limb is bilabiate, with the upper lip being slightly galeate and bifurcated, and both petals are reflexed. The lower lip is trilobate, with the middle lobe slightly larger, both obovate, rounded at the terminus, and marginally spreading (Figure 1b).

The morphology of *M. savatieri* flowers includes four stamens, of which two are more prominent and inserted on the floral tube. The anterior pair reaches a length of about 7 mm, while the posterior pair measures about 6 mm. The anthers are yellowish, dorsifixed, and slightly exposed within the corolla’s throat. They are two-loculed, parallel, and equal, measuring 0.15–0.30 cm in length and 0.06–0.08 cm in width, and are distinctly separated. The anthers are long and ovate, tapering at the lower section, with a small convex tip and a longitudinal cleft (Figure 1g).

The ovary is long and ovate with a slender, elongated style that has a curved tip. The stigma is oblong and covered with white tomentum (Figure 1f). The growth of *M. savatieri* outdoors is more influenced by rainfall, leading to a smaller overall plant size compared to indoor-grown specimens; however, the single-flower morphology remains consistent between the two environments.

#### 2.1.2. Dynamics of Flowering in *M. savatieri*

The study observed no significant differences in the blooming dynamics of individual flowers and inflorescences of *M. savatieri* under greenhouse and outdoor field cultivation. Differences in the flowering period, flower count, and fruiting rate were, however, noted between whole plants and populations under varying cultivation practices.

The racemes of *M. savatieri* are terminal, with single flowers arranged oppositely. The number of flowers per flowering branch ranged from 2 to 10, depending on branch length and plant growth conditions, with an average of 6.33 ± 2.36 flowers. The flowering period lasted about 20.89 ± 7.03 days. Inflorescences bloomed sequentially from the base upwards, with typically one to two flowers opening simultaneously per inflorescence (Figure 2). Additionally, branches sprouting between March and April were capable of flowering within the same year.

The floral morphology of *M. savatieri* exhibited notable changes during its developmental stages, which were categorized into four phases: the bud stage (1–3 days), bud appearance (4–8 days), pollination stage (9–14 days), and fruiting stage (15–35 days), as shown in Figure 3. Flower development progressed from bud to bloom in about five days. Three days after blooming, the petals began to darken and crumple, with petals wilting within five to six days. By the ninth day, ovary expansion began, and fruit dehiscence occurred around the twentieth day. The flowering of *M. savatieri* predominantly occurred in the early morning hours, specifically between 4:00 and 6:00 AM (Figure 4). Under greenhouse cultivation, flowering began in late February. The blooming phase commenced in late March, with most plants completing their flowering cycle by mid-May. The overall flowering period concluded in early June (Figure 5). The number of flowers per plant ranged from 17 to 241, with an average of 60.63 ± 39.94 flowers and a fruiting rate of 87.10 ± 8.67%. Each capsule contained 72.57 ± 16.54 seeds. The duration from the first flower opening to the full bloom of the entire plant was about 7 days. The full bloom period lasted about 7 days, followed by the sequential opening of a few flowers under favorable conditions. The flowering period for the whole plant ranged from 30 to 60 days.

In outdoor field cultivation, *M. savatieri* initiated flowering in late March. The blooming phase extended from late March to late April, concluding by the end of April (Figure 6). Under optimal conditions characterized by sunny weather and minimal rainfall, the number of flowers per plant varied between 26 and 139, averaging 58.32 ± 29.92 flowers, with a fruiting rate of 78.27 ± 10.01%. Under rainy conditions, the average flower count per plant was 48.00 ± 19.61, with a fruiting rate of 56.67 ± 16.37%. The time from the first bloom to the full flowering of the plant spanned about 3 to 4 days, with the full bloom phase lasting about 7 days. A limited number of flowers continued to open sequentially under favorable conditions. The overall flowering duration for the entire plant ranged from 20 to 30 days.

#### 2.1.3. *M. savatieri* Loose Powder Dynamics

The dynamics of stamen development were consistent under both indoor and outdoor cultivation conditions for *M. savatieri*. Anther development progressed in parallel with flower bud growth. At the initial stage, when flower buds emerged, stamens were clustered together, and the anthers displayed a two-locular structure. About three days before anthesis, flower buds slightly enlarged, with the anthers retaining their two-locular form. One day prior to anthesis, a clear separation into two chambers was observed, and the anthers became fully developed (Figure 7(c3)).

Within one hour after corolla blooming, the anthers initiated dehiscence and pollen dispersal (Figure 7(d3)), with the anterior pair of stamens dehiscing first. After blooming, the anthers adhered to the upper lip of the corolla and slightly protruded at the corolla’s throat. The pollen appeared white, retained its color over time, and was completely dispersed upon anther dehiscence.

The pollen dispersal of *M. savatieri* began about two hours after flowering, with the stamens releasing pollen sequentially. Under sunny conditions, pollen dispersal was concentrated between 7:00 and 11:00 AM, with the majority of pollen released in the morning (refer to Figure 7(e2)). Elevated air humidity inhibited pollen dispersal, leading to a prolonged release on overcast days. In such cases, pollen occasionally persisted for 2 to 3 days after anther dehiscence.

#### 2.1.4. The Development of Stigma in the *M. savatieri*

The initial blooming phase of *M. savatieri* is marked by the position of the stigma near the upper lip, with its tip extending beyond the corolla and curving downward. Within two days, the stigma adjusts its position to match the flower’s depth (Figure 8). This dynamic movement of the pistil promotes cross-pollination during the early flowering phase, while self-pollination occurs if external pollen is unavailable.

Prior to pollination, the stigma appeared white and was covered with downy hairs. It was capable of receiving pollen immediately following anthesis. The stigma retained its white and downy appearance one day before and after corolla dehiscence, a phase during which its activity peaked. During this period, the pollination and fruiting rate reached 93.30 ± 9.43%.

The stigma maintained optimal activity for 2 to 3 days post-anthesis. During this phase, the style began to redden from the base, and the stigma turned yellow, resulting in a pollination and fruiting rate of 80.00 ± 16.33%. By 4 to 5 days post-anthesis, the style had fully reddened, and the stigma transitioned to a yellow-brown color. Stigma activity significantly declined at this stage, leading to a reduced fruiting rate of 25.00 ± 5.00%. By 5 to 7 days post-anthesis, the stigma had desiccated and become inactive (Figure 9).

Manual pollination of newly opened flowers revealed differences in stigma morphology between pollinated and unpollinated flowers. Contact with pollen caused the downy hairs on the stigma to decrease rapidly, and the stigma turned yellow within one hour (Figure 10).

### 2.2. The Self-Compatibility of M. savatieri

The study demonstrated that *M. savatieri*, when subjected to artificial emasculation and bagging, failed to produce fruit, resulting in a D/A ratio of 0. This result indicates that *M. savatieri* is unable to produce seeds through apomixis and relies on pollination for reproduction.

The fruiting rate of indoor *M. savatieri* subjected to artificial emasculation without bagging (pure crossbreeding) was observed to be 32.50%, whereas the rate under outdoor conditions was lower at 15.63%. In contrast, the natural fruiting rate of indoor *M. savatieri* natural pollination (control group, CK) was 88.57%, while outdoor conditions yielded a lower rate of 56.67%. Additionally, artificial pollination experiments revealed that artificial self-pollination resulted in a fruiting rate of 85.29%, while artificial cross-pollination achieved a higher fruiting rate of 90.00% (Table 1). These findings suggest that the pollination success rate in outdoor *M. savatieri* is reduced, primarily due to the overlap of the flowering period with the rainy season in its native region, which results in increased rainfall during this critical phase.

Analysis of the outcrossing rate revealed that indoor-cultivated *M. savatieri* achieved a rate of 28.80%, whereas outdoor-cultivated plants exhibited a lower rate of 9.60%. This indicates a higher occurrence of heterogamy under indoor conditions compared to outdoor environments. The outcrossing rate for both conditions ranged between 5% and 50%, suggesting that *M. savatieri* is typically an often cross-pollinated plant, capable of both self-pollination and outcrossing pollination.

Floral structures of *M. savatieri* exhibited monoecy with herkogamy (stigma–anther separation) and synchronized anther dehiscence with stigma receptivity. An outcrossing index (OCI) of 3 (Dafni’s criterion) indicated morphological adaptations favoring cross-pollination, despite the retained self-compatibility.

The A1 group underwent natural pollination in a greenhouse environment (CK1), while the A2 group experienced natural pollination in a field environment (CK2). The B group was directly bagged. The C1 group involved emasculation without bagging in the greenhouse, and the C2 group involved emasculation without bagging in the field. The D group featured emasculation followed by bagging. The E group focused on artificial geitonogamy (self-pollination within the same flower), and the F group engaged in artificial xenogamy.

### 2.3. Determination of Pollen Viability of M. savatieri

The analysis of various staining techniques for *M. savatieri* pollen revealed that scattered pollen remained colorless when treated with TTC (2, 3, 5-Triphenyltetramlium chloride) solution. However, superior staining results were obtained when entire pollen sacs were immersed in the solution (Figure 11b,c). This suggests that the respiration activity of *M. savatieri* pollen is relatively weak, and the TTC solution undergoes a detectable color change only when densely packed pollen is involved in respiration. The I_2_-KI method was found unsuitable for assessing pollen viability in *M. savatieri* due to the pollen’s yellow-brown color and the absence of the expected blue coloration (Figure 11d). Acetocarmine effectively stained the nuclei of viable *M. savatieri* pollen cells red (Figure 11e). The MTT (3-(4,5)-dimethylthiahiazo (-z-y1)-3,5-di-phenytetrazoliumromide) assay produced a blue-violet crystalline substance through the action of dehydrogenase enzymes in viable pollen grains, staining viable pollen blue, while non-viable pollen displayed a distinct yellowish-brown color (Figure 11f). The FDA (fluorescein diacetate) staining method revealed fluorescence in viable *M. savatieri* pollen under blue light (Figure 11h). Three staining methods—acetocarmine, MTT, and FDA—were employed to assess the viability of antler grass pollen post-dispersal. The acetocarmine method demonstrated that pollen viability exceeded 95% within three days of dispersal and maintained high activity over an extended duration. The MTT method indicated a viability of 92.76% ± 2.98% after three days, with pollen activity recorded at 83.37% ± 4.12% after 60 days. The FDA method further confirmed that pollen viability remained above 95% within the initial three days, with sustained high activity over time (Figure 12).

## 3. Discussion

### 3.1. Stigma Dynamics and Pollination Characteristics of M. savatieri

The floral and pollination characteristics, along with breeding systems, demonstrate mutual adaptation in plants. *M. savatieri* is characterized by a predominantly white to light pink corolla, with occasional dark pink variations. The corolla diameter exceeds 1 cm, and the lower labellum features a bright yellow color block, which effectively attracts pollinators.

Approach herkogamy, the most common form of herkogamy, involves pollinators entering the flower and first contacting the highly protruding stigma, which may also intercept pollen carried by wind [19]. The style of *M. savatieri* extends beyond the corolla, while the stamens remain close to the upper lip, consistent with stigma-probing androgynous ectopia.

Delayed self-fertilization is a widely recognized reproductive adaptation that ensures seed production in the absence of pollinators while prioritizing outcrossing when pollinators are abundant. The curling of the style, which brings the stigma surface into contact with the anthers or fallen pollen on non-stigma regions during late anthesis, is a typical floral movement facilitating delayed self-pollination [20].

Similar floral behavior is observed in mallow plants, such as *Hibiscus trionum* L., where flowers feature prominent stigmas capable of bending in the opposite direction shortly after opening. This movement enables the stigma to contact anthers, promoting self-pollination. If stigmas receive pollen before or during the bending process, self-pollination is inhibited, as the stigma either remains in or reverts to an upright position [21]. In *Viola pubescens*, delayed self-fertilization occurs when the stigma moves downward to contact pollen that has fallen onto anterior petals from the anthers [22]. In *M. savatieri*, the style’s downward bend allows the stigma to make contact with pollen on the lower lip or that remaining in the anthers, promoting delayed self-fertilization.

Reproductive mechanisms represent the primary link between progenitors and their descendants and among descendants within advanced plants and animals. Unlike animals, reproductive strategies in advanced plants exhibit considerable variation across taxa, as well as among groups and individuals within a single species. Understanding the reproductive system is essential for investigating gene transfer within plant populations.

The outcrossing rate is a key characteristic of the mating system. Heterogamous plants typically exhibit an outcrossing rate exceeding 50%, while normally heterogamous plants demonstrate an outcrossing rate between 5% and 50%. In contrast, self-pollinated plants generally show an outcrossing rate below 5%. Factors such as plant traits and cultivation conditions influence the natural hybridization rate. For example, increasing the distance between Setaria plantings reduces the likelihood of outcrossing, whereas strong winds during the flowering period, particularly in the wind’s direction, enhance the probability of natural hybridization.

The experimental results confirmed that *M. savatieri* functions as a normally heterogamous pollinator under both indoor and outdoor cultivation conditions. Observations of the floral structures of *M. savatieri* revealed its monoecious characteristics, with stamens and pistils of unequal lengths. The synchronization of anther pollination with peak stigma activity, combined with an outcrossing index (OCI) of 3 based on Dafni’s criterion, suggests that *M. savatieri* exhibits traits characteristic of consistently heterogamous plants.

### 3.2. Flowering Time, Stigma Activity, and Suitable Conditions for Artificial Pollination of M. savatieri

Among flowering plants, hybridization is a key strategy employed by breeders to develop progeny with advantageous agronomic traits or enhanced resistance to abiotic and biotic stresses. This process is typically facilitated through artificial pollination techniques.

The survival duration of pollen and the acceptance period of stigmas are closely associated with species-specific characteristics. *M. savatieri* exhibits a raceme inflorescence, with floral development divided into four distinct phases: the bud stage (1–3 days), bud appearance (4–8 days), pollination stage (9–14 days), and fruiting stage (15–35 days). A single flowering branch typically bears 2 to 10 flowers, with an average bloom count of 6.33 ± 2.36. The complete flowering duration of a branch is about 20.89 ± 7.03 days. Under indoor cultivation, *M. savatieri* demonstrated an extended flowering period from late February to mid-May, with individual plants flowering for 30 to 60 days. In contrast, outdoor cultivation resulted in a shorter flowering period from late March to mid-May, with individual plants flowering for 20 to 30 days.

*M. savatieri* exhibits strong pollen vitality one day prior to anthesis. Under natural conditions, anthers begin to open one hour after flower blooming, with pollen dispersal completed within five hours under sunny conditions. In cloudy or rainy weather, the pollen dispersal period extends to 2–3 days. Notably, pollen vitality remains high for up to 60 days post-dispersal.

The findings indicate that optimal pollination in *M. savatieri* occurs within specific time windows relative to flower opening. One day prior to anthesis, stigma activity peaks, achieving a pollination and fruiting rate of 93.30 ± 9.43%. Stigma activity remains high during the first bloom. Between 2 and 3 days post-flowering, the stigma remains favorable for pollination, with the style beginning to redden from the base and the stigma turning yellow. During this phase, the pollination and fruiting rate is 80.00 ± 16.33%. By 4 to 5 days post-flowering, the style becomes fully red, and the stigma transitions to a yellowish-brown color, with activity significantly reduced, leading to a fruiting rate of 25.00 ± 5.00%. After 5 to 7 days, the stigma desiccates and becomes inactive. Based on these observations, it is recommended that artificial pollination be conducted within the first 1 to 2 days following flower opening to maximize pollination success and fruit set.

### 3.3. Staining Methods for the Detection of Suitable Viability of Pollen from M. savatieri

During the experimental period, no significant changes were observed in the appearance of pollen, indicating that evaluating pollen viability based solely on its physical appearance is not feasible. The primary methods for assessing pollen viability include in vitro germination, pollination, and staining. The in vitro germination method, although accurate, is complex, time-consuming, and requires optimization of conditions, making it impractical for rapid viability detection. The pollination method, on the other hand, is highly influenced by environmental factors and has been employed less frequently in recent years. Staining techniques are more widely used due to their simplicity, rapidity, and effectiveness.

The I_2_-KI staining method is based on the accumulation of starch in pollen, with blue-stained pollen considered viable [23]. In the present study, the pollen of *M. savatieri* was stained yellowish-brown with I_2_-KI, indicating that this method is unsuitable for assessing pollen viability, likely due to the low starch content in its pollen.

The TTC (2, 3, 5-Triphenyltetramlium chloride) method, which induces a red coloration through a reaction with succinate dehydrogenase in viable cells, is widely used for pollen viability assessment. In the case of *M. savatieri*, dispersed pollen remained colorless after treatment with the TTC solution. However, better staining results were achieved by immersing the entire pollen sac in the TTC solution. This may be attributed to the weak respiration of *M. savatieri* pollen, as the dense aggregation of pollen is required to generate sufficient respiration to induce a detectable color change in TTC.

Acetocarmine staining targets the nuclei of pollen cells and has demonstrated clear and distinguishable results in species. In the case of *M. savatieri*, ace to carmine staining resulted in rapid and distinct staining, but the viability results remained consistently above 95% over extended durations, suggesting potential overestimation of viability.

MTT staining functions by utilizing the dehydrogenase enzyme in active pollen, which reduces MTT to a water-insoluble blue-violet precipitate. *M. savatieri* pollen exhibited significant color differentiation with the MTT staining solution, making it an effective technique for assessing pollen viability.

The FDA fluorescence staining method relies on fluorescein diacetate (FDA) reacting with pollen grain esterases to produce fluorescein, thereby enabling staining and viability assessment. However, it has a tendency to overestimate pollen viability. *M. savatieri* pollen, stained using the FDA method, emitted strong green fluorescence, with results remaining above 95% for extended periods, indicating possible overestimation when applying this technique for viability assessment.

Among the five pollen staining methods evaluated in this study, I_2_-KI and TTC staining proved ineffective for assessing *M. savatieri* pollen viability. Acetocarmine and FDA staining methods produced reliable yet potentially elevated results. MTT staining emerged as the most suitable and accurate method for assessing *M. savatieri* pollen viability due to its clear and consistent differentiation between viable and non-viable pollen. The pollen viability of *M. savatieri* exceeded 85% in both indoor and outdoor populations, as determined by MTT staining. This high viability ensures a sufficient supply of functional pollen grains, even under outdoor environmental stressors such as heavy rainfall.

## 4. Materials and Methods

### 4.1. Materials

The observation samples were planted at the *M. savatieri* breeding base in Yichun City, Jiangxi Province, China. Both indoor and outdoor cultivated plants were utilized, with seeding carried out in May 2022 and transplantation conducted in April 2023. Flower buds exhibiting normal growth and approaching bloom were selected for study over two consecutive flowering periods (April to June) in 2023 and 2024. *Gardenia jasminoides* was used as a host for *M. savatieri* during the co-cultivation of seedlings. For indoor cultivation, *M. savatieri* seedlings were transplanted at intervals of 30 cm between plants and rows within plastic greenhouses, which were ventilated by rolling up the bottom sides to a height of 1 m on sunny days. For outdoor cultivation, seedlings were transplanted into open fields, maintaining the same spacing as in indoor conditions.

### 4.2. Observational Analysis of Floral Characteristics and Blooming Patterns

#### 4.2.1. Analysis of Floral Appearance and Blooming Dynamics

In May 2024, a study was conducted by selecting 100 flowering branches from 50 *M. savatieri* plants to monitor and document their flowering and fruiting processes. Plant morphology was assessed using a scale, while the morphology of organs was photographed and measured with a stereo microscope (OLYMPUS SZX-ILLB2-200, Tokyo, Japan) and a camera.

#### 4.2.2. Analysis of Pollen Dynamics

Ten randomly selected flowering branches of *M. savatieri*, poised to bloom in May 2023 and May 2024, were monitored to assess pollen dispersal. Observations were conducted to record variations in anther dispersal patterns.

#### 4.2.3. Analysis of the Process of Stigma Development

In May 2024, a study was performed on 184 flowers from 154 *M. savatieri* plants. Flowers nearing bloom underwent a procedure where male components were removed, and stigmas were enclosed in 6 cm × 10 cm sulfate paper bags starting from the second day. Observations of stigma morphology and continuous pollination were carried out, with detailed records of stigma characteristics at pollination. The fruiting rate was also examined at seed maturity. Furthermore, daily samples of stigmas were collected from pre-blooming flowers to evaluate pollinability using the benzidine–hydrogen peroxide method.

### 4.3. Analysis of Floral Self-Compatibility

#### 4.3.1. Determination of Outcrossing Index (OCI) Values

The Outcrossing index (OCI) values were determined following Dafni’s criteria to analyze the breeding system [24]. The methodology included: (1) recording the flower or inflorescence diameter as 0 for <1 mm, 1 for 1–2 mm, 2 for 2–6 mm, and 3 for >6 mm; (2) documenting the time interval between anther dehiscence and stigma pollination as 0 if the stigma matures first and 1 if the anther matures first; (3) noting the spatial position of the stigma and anther as 0 for the same height and 1 for spatial separation. The sum of these three parameters was used to determine the OCI value. The criteria for classification are as follows: OCI = 0 indicates a closed fertilization system; OCI = 1 signifies exclusive self-fertilization; OCI = 2 denotes parthenogenetic self-fertilization; OCI = 3 represents self-fertilization affinity, occasionally involving pollinators; OCI = 4 suggests partial self-fertilization affinity with heterogamy requiring pollinators; OCI = 5 indicates exclusive heterogamy.

#### 4.3.2. Analysis of the Rate of Natural Outcrossing

In May 2024, a total of 189 flower buds nearing bloom were selected from 75 *M. savatieri* plants for testing under six distinct treatments. These treatments included: 35 flower buds subjected to natural pollination in a greenhouse environment (A1); 23 flower buds subjected to natural pollination in a field environment (A2); 30 flower buds subjected to directly bagged (B); 40 flower buds subjected to emasculation without bagging in the greenhouse (C1); 31 flower buds subjected to emasculation without bagging in the field (C2); and 30 flower buds subjected to emasculation and bagged (D); 34 flower buds subjected to artificial geitonogamy (E); 30 flower buds subjected to artificial xenogamy (F). Bagging is performed in 6 cm × 10 cm sulfate paper bags, with each flower individually segregated in a bag. The study primarily examined the fruiting rate. The ability of *M. savatieri* to reproduce through apomixis was evaluated by calculating the ratio of D/A. Furthermore, the outcrossing rate was determined using the formula: (A1/(B + C1)) × C1 × 100% for the greenhouse environment and (A2/(B + C2)) × C2 × 100% for the field environment, based on the fruit set rate for each treatment.

### 4.4. Determination of Pollen Viability

In May 2024, fresh pollen samples were collected and placed in centrifuge tubes. Multiple staining techniques, including I_2_-KI, TTC, MTT, Acetocarmine, and FDA, were applied to evaluate pollen viability. The aim was to identify the most effective method for determining pollen viability.

The I_2_-KI staining method is based on the accumulation of starch in pollen, with blue-stained pollen considered viable [23].

The TTC (2, 3, 5-Triphenyltetramlium chloride) method, which induces a red coloration through a reaction with succinate dehydrogenase in viable cells, is widely used for pollen viability assessment [25].

Acetocarmine staining targets the nuclei of pollen cells and stains viable pollen red, demonstrating clear and distinguishable results in species such as peanut [26].

MTT (3-(4,5)-dimethylthiahiazo (-z-y1)-3,5-di-phenytetrazoliumromide) is a yellow-colored dye. MTT colorimetric assay can detect cell survival and growth. The principle of the assay is that the enzyme succinate dehydrogenase in the mitochondria of living cells reduces exogenous MTT to the water-insoluble blue-violet crystalline formazan and deposits it in the cells, whereas dead cells do not have this function [27].

The FDA (fluorescein diacetate) reacts with pollen grain esterases to produce fluorescein, thereby enabling staining and viability assessment [25].

## 5. Conclusions

*M. savatieri* is classified as a cross-pollinated post-flower species with monoecious flowers. Indoor cultivation is recommended for this plant, as environmental factors such as rainfall during flowering significantly reduce fruiting rates and challenge centralized harvesting in outdoor environments due to sequential fruit ripening and water-induced capsule dehiscence.

For pollen viability assessment, the MTT staining method was found to be more suitable than I_2_-KI or TTC staining. As a frequently heterogamous pollinated plant, systematic crossbreeding strategies should be implemented. However, to enhance breeding outcomes, parental plants must undergo rigorous self-crossing selection to eliminate inferior individuals and retain superior pure lines as hybrid parents.

## Figures and Tables

**Figure 1 plants-14-00715-f001:**
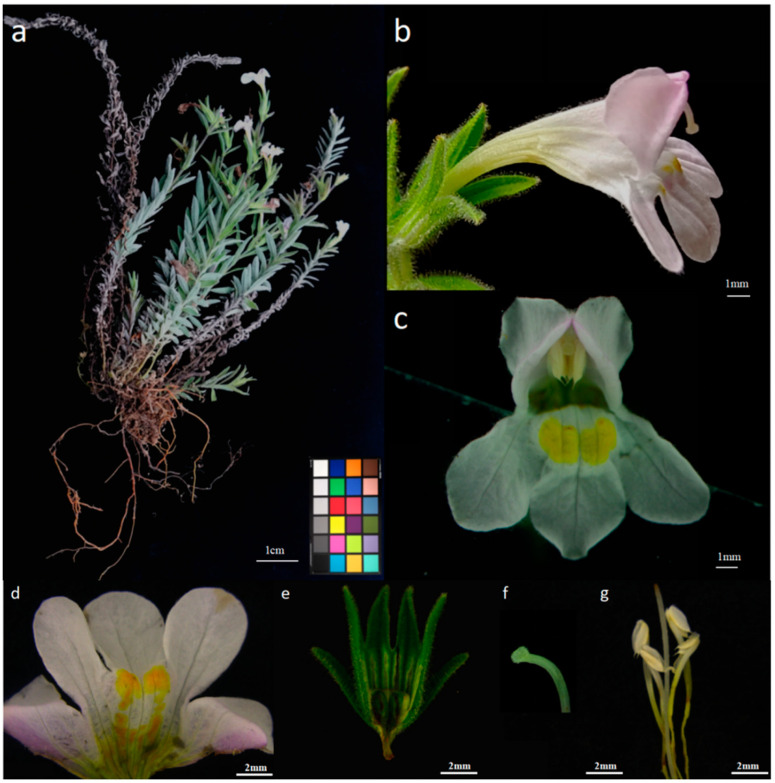
Flowering plant, single flower morphology, and flower organs of *M. savatieri* (*Monochasma savatieri* Franch. ex Maxim.) (**a**): flowering plant; (**b**): side view of flower; (**c**): front view of flower; (**d**): petal; (**e**): calyx; (**f**): stigma; (**g**): stamens.

**Figure 2 plants-14-00715-f002:**
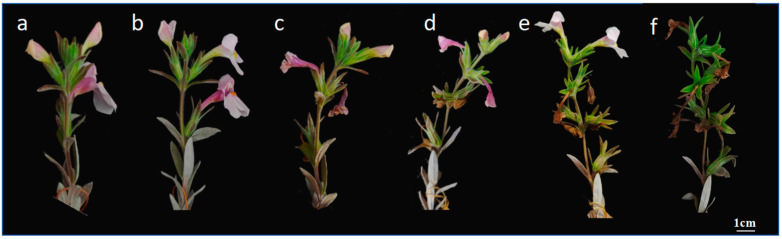
Dynamics of inflorescence development in *M. savatieri*: (**a**): 17 March; (**b**): 18 March; (**c**): 23 March; (**d**): 28 March; (**e**): 30 March; (**f**): 7 April.

**Figure 3 plants-14-00715-f003:**
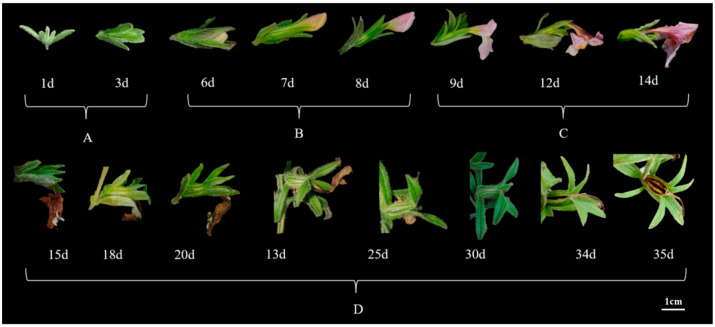
Flower morphology of *M. savatieri* at different developmental stages: (**A**): flower bud stage; (**B**): present bud stage; (**C**): pollination stage; (**D**): fruiting stage.

**Figure 4 plants-14-00715-f004:**
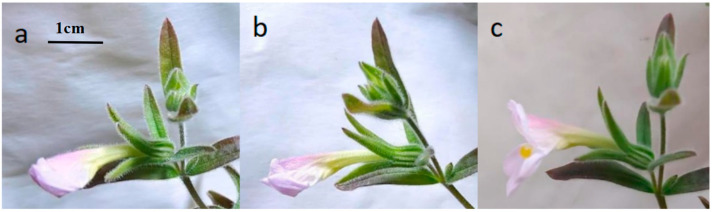
Dynamics of flower opening in *M. savatieri*: (**a**): 02:30 a.m.; (**b**): 04:30 a.m.; (**c**): 06:00 a.m.

**Figure 5 plants-14-00715-f005:**
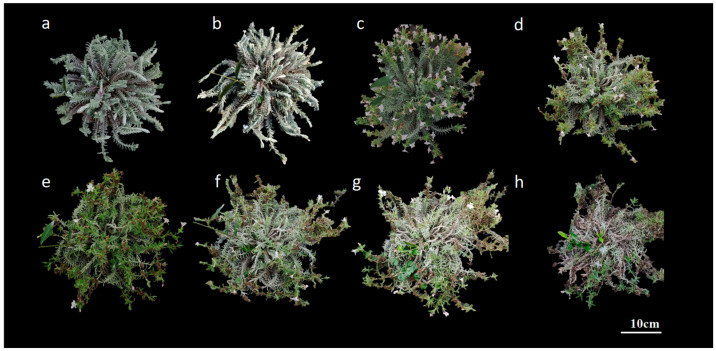
Dynamics of whole plant development of *M. savatieri* cultivated in greenhouse: (**a**): 25 March; (**b**): 28 March; (**c**): 10 April; (**d**): 20 April; (**e**): 26 April; (**f**): 30 April; (**g**): 13 May; (**h**): 30 May.

**Figure 6 plants-14-00715-f006:**
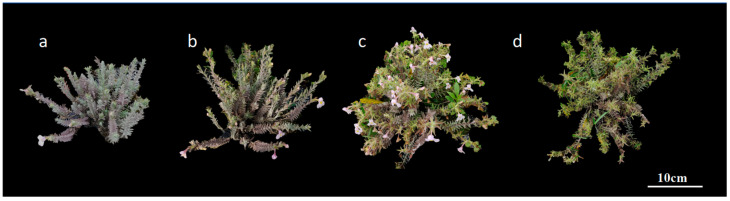
Dynamics of whole plant development in outdoor field cultivation of *M. savatieri*: (**a**): 25 March; (**b**): 28 March; (**c**): 14 April; (**d**): 22 April.

**Figure 7 plants-14-00715-f007:**
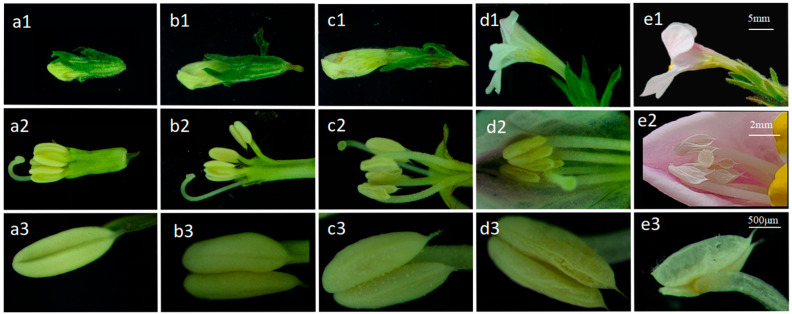
Anthers at various developmental stages: (**a1**–**a3**) represent the initial appearance of buds, with anthers exhibiting a 2-chambered morphology; (**b1**–**b3**) show slightly larger buds, maintaining the 2-chambered morphology; (**c1**–**c3**) depict flowers on the verge of opening, with fully developed anthers that will dehisce upon external force; (**d1**–**d3**) indicate the initial blooming phase of flowers, where anthers naturally dehisce; (**e1**–**e3**) correspond to the stage 5 h post-flower opening, where anthers have completed pollen dispersal.

**Figure 8 plants-14-00715-f008:**
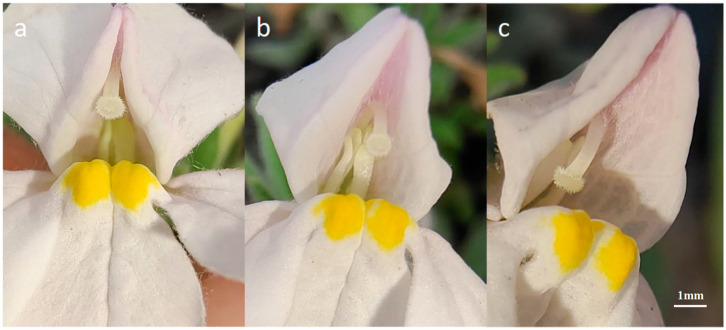
Stigma curvature following the blooming of *M. savatieri* flowers: (**a**): 1 d post-bloom; (**b**): 2 d post-bloom; (**c**): 3 d post-bloom.

**Figure 9 plants-14-00715-f009:**
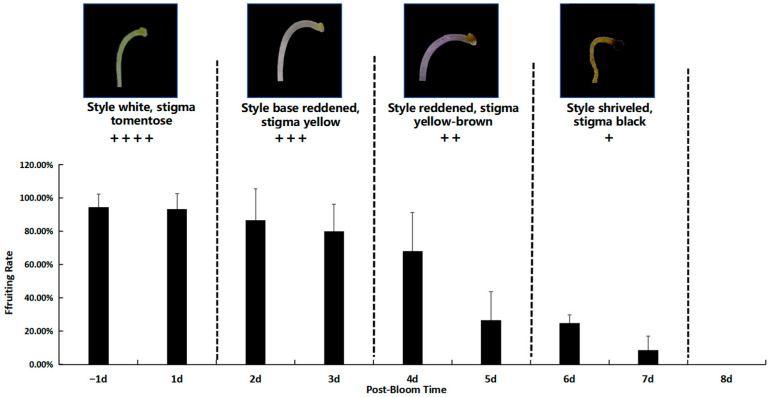
The correlation between stigma activity and the flowering period of *M. savatieri*: +: Stigma with receptivity; ++/+++: Stigma with moderately strong receptivity; ++++: Stigma with strong receptivity.

**Figure 10 plants-14-00715-f010:**
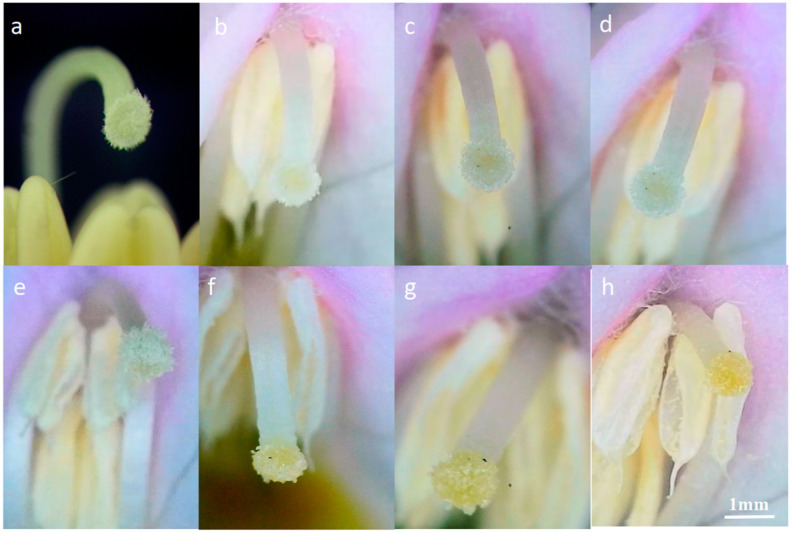
Stigma development was observed under a microscope at various time intervals: (**a**): 0 h post-anthesis without pollination; (**b**): 1 h post-anthesis without pollination; (**c**): 4 h post-anthesis without pollination; (**d**):12 h post-anthesis without pollination; (**e**): 0 h post-pollination; (**f**): 1 h post-pollination; (**g**): 4 h post-pollination; (**h**): 12 h post-pollination.

**Figure 11 plants-14-00715-f011:**
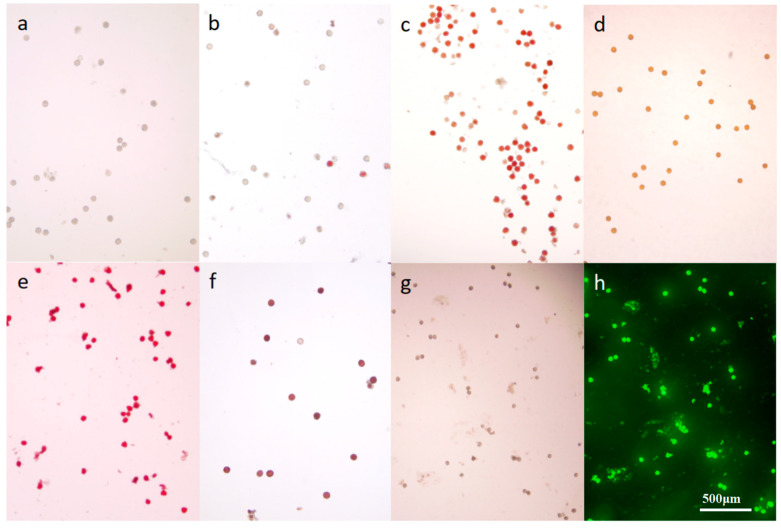
Staining of pollen by different methods (**a**): Pollen grains under water; (**b**): TTC-stained pollen grains; (**c**): TTC-stained pollen grains dispersed from pollen sacs; (**d**): I_2_-KI-stained pollen grains; (**e**): Acetocarmine-stained pollen grains; (**f**): MTT-stained pollen grains; (**g**): FDA-stained pollen grains viewed under white light; (**h**): FDA-stained pollen grains viewed under blue light.

**Figure 12 plants-14-00715-f012:**
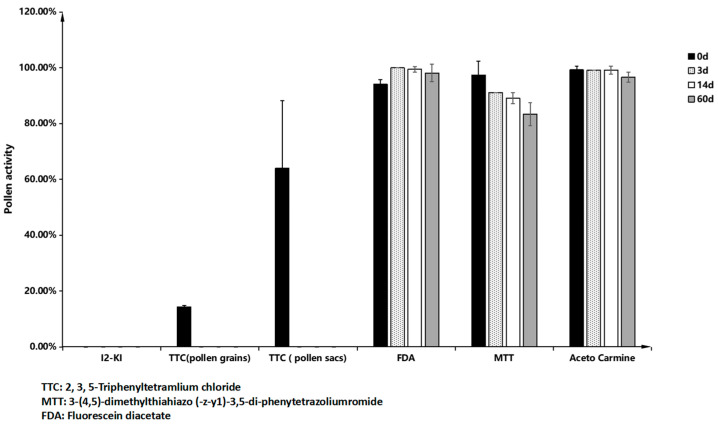
Pollen activity detected by different staining methods at 0 d, 3 d, 14 d, and 60 d after pollen dispersal.

**Table 1 plants-14-00715-t001:** Fruiting rate of *M. savatieri* in different treatments.

Pollination TEST	Number of Flowers	Number of Fruits Fruiting	Fruiting Rate %
A1	35	31	88.57%
A2	24	14	56.67%
B	30	23	76.67%
C1	40	13	32.50%
C2	31	5	15.63%
D	30	0	0
E	34	29	85.29%
F	30	27	90.00%

## Data Availability

The original contributions presented in the study are included in the article, further inquiries can be directed to the corresponding author.

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
