# Peer review of "Study on Flowering Dynamics and Pollination Habits of Monochasma savatieri Under Artificial Cultivation Conditions"

_plants, 2025, doi:10.3390/plants14050715_

Round 1

Reviewer 1 Report

Comments and Suggestions for Authors

I really appreciate this article. In it, Haoqian Zhang, Qiuling Wang and Jianhe Wei make a detailed study of the floral biology of such an interesting species, especially from a pharmacological point of view, as Monochasma savatieri. The original approach of the work is accompanied by a qualified and focused experimental design. The analysis of the floral biology is exhaustive and provides results of great importance. The statistical treatment is appropriate. The graphs clearly illustrate what needs to be emphasised. The discussion and conclusions are well developed and correspond to the results.

Consequently, I consider it adequate to publish the article without modification.

Author Response

Dear Reviewer,

We sincerely appreciate your thoughtful review and valuable comments on our manuscript titled "Study on flowering dynamics and pollination habit of Monochasma savatieri under artificial cultivation conditions." Your thorough evaluation and positive feedback have greatly encouraged us and are highly appreciated.

We are particularly grateful for your recognition of the originality of our work and the well-designed experimental approach. As you noted, we have made significant efforts to conduct a comprehensive analysis of the floral biology of this fascinating species. The detailed experiments, statistical treatments, and clear graphical illustrations were all aimed at ensuring the reliability and scientific rigor of our findings. Your acknowledgment of these aspects is truly inspiring.

We are also deeply appreciative of your compliments regarding the discussion and conclusions. Your validation of our work’s significance and potential practical applications is both motivating and reassuring.

It is a great honor to receive such positive feedback from an expert like you. We will continue to strive for excellence in our research, aiming to contribute more meaningful insights to the scientific community.

Thank you once again for your time, effort, and constructive comments. We look forward to your continued guidance and support.

Best regards,
Haoqian Zhang, Qiuling Wang, and Jianhe Wei
Research Team

Reviewer 2 Report

Comments and Suggestions for Authors

Comments and Suggestions for Authors

The study aimed to investigate the reproductive biology of Monochasma savatieri Franch. ex Maxim., focusing on its floral structure, developmental stages, flowering dynamics, pollen dispersal characteristics, stigma development and longevity, as well as self-crossing affinity in both indoor and outdoor cultivation conditions. The findings hold significant value for the selection and breeding of M. savatieri.

 The manuscript is constructed according to requirements of “Plants”. The applied research methods are suitable for achieving the research objective but their description is incomplete /especialy the methods for pollen viability estimation/. The results are well illustrated.

Additional remarks and suggestions can be made:

Abstract:

-          In line 21to be provided the meaning of OCI.

Introduction

-          In the first sentence /line 29-30/ add the family name /after the name of the genus/.

-          The part on pharmaceutical uses and activity of active compounds of M. savatieri is too long and should be shortened as it is not in the focus of the study.

-          In this section is recommendable to provide info about flowering dynamics and pollen dispersal characteristics (if exist) for species close to the studied species (from the same genus, tribe, family)

 Discussion

-In point 3.3 Staining methods for the detection of suitable viability of pollen from M. savatieri, the description of staining methods for assessing pollen viability /lines 364-365, 374-375, 393-394, 399-401/ should be moved to “Materials and Methods”, section 4.4. Determination of pollen viability. Also, examples of the application of these staining methods to other species are unnecessary.

- Add a comment on the established high pollen viability and its importance for the success of outdoor cultivation by ensuring a sufficient amount of viable pollen grains, thus ensuring successful pollination.

Conclusions

 -          This section should be shortened. All parts provide data suitable for the “Results”. Only the last part /lines 499-504/ and final sentences of other parts /lines 483-485, 493-494 /from “…indoor cultivation ..”, 497-498, from “..the MTT staining method was found …”can be used for this section.

- In this section shoult be incuded the findings about flowers characteristics of M. savatieri: M. savatieri. is classified as a cross-pollinated post-flower-species with monoecious flowers.

In conclusion, this manuscript is recommended for publication in "Plants" after consideration of the comments made.

Author Response

Comments 1: Abstract:

-  In line 21to be provided the meaning of OCI.

Response 1: Thank you for pointing this out. We agree with this comment. Therefore, we  explicitly define outcrossing index (OCI)  in line 21. And at Materials and Method stage, relevant references were added.

line 21“[The plant, with an outcrossing index (OCI) of 3, ]”

Comments 2: -  In the first sentence /line 29-30/ add the family name /after the name of the genus/.

Response 2: Thank you for pointing this out. We will add the family name, Scrophulariaceae, after the genus name Monochasma to provide the full taxonomic classification of M. savatieri.

line 30“[Monochasma savatieri Franch. ex Maxim. is a semi-parasitic perennial herb belongs to the Scrophulariaceae family. ]”

Comments 3: L. 70 to 113, please add at least some references to Fig 1.

L. 99 to 113. Diagrams of the different floral organs (or photos of floral dissections) would be interesting to include to illustrate this part.

Response 3: Thank you for your suggestion. We have revised the manuscript to address this point. To streamline the introduction, we will shorten the section on pharmaceutical uses and active compounds, ensuring it remains concise and relevant to the study's objectives.

Line34-38“[M. savatieri, a traditional Chinese medicinal herb, is used holistically to treat conditions such as respiratory infections, inflammatory disorders, and gynecological ailments [1]. Its key bioactive components—flavonoids, alkaloids, saponins, and polysaccharides [2,5]—contribute to broad pharmacological activities, including antimicrobial, antioxidant, and anti-inflammatory effects [4,6–15]. Recent studies emphasize its anticancer potential [16].]”

Comments 4: In this section is recommendable to provide info about flowering dynamics and pollen dispersal characteristics (if exist) for species close to the studied species (from the same genus, tribe, family)

Response 4: Agree. We have incorporated your comments into the revised version. Added research about flowering dynamics and pollen dispersal characteristics of the same family.

line51-54“[In the study of the Scrophulariaceae family, the reproductive system of Scrophularia ningpoensis Hemsl is partially self-compatible, requiring pollinators for outcrossing with a natural outcrossing rate of 72.15%, classifying it as a typical outcrossing plant[16]]”

Comments 5: -In point 3.3 Staining methods for the detection of suitable viability of pollen from M. savatieri, the description of staining methods for assessing pollen viability /lines 364-365, 374-375, 393-394, 399-401/ should be moved to “Materials and Methods”, section 4.4. Determination of pollen viability. Also, examples of the application of these staining methods to other species are unnecessary.

Response 5: Thank you for your suggestion.In response to your suggestion, we have carefully reviewed the relevant sections. The descriptions of the staining methods for assessing pollen viability, as mentioned in lines 364-365, 374-375, 393-394, and 399-401, have been moved to the "Materials and Methods" section under "4.4. Determination of Pollen Viability." This adjustment ensures that the discussion section remains focused on interpreting the results rather than detailing the methods.

Line476-489“[4.4. Determination of Pollen Viability.]"

Comments 6: - Add a comment on the established high pollen viability and its importance for the success of outdoor cultivation by ensuring a sufficient amount of viable pollen grains, thus ensuring successful pollination.

Response 6: Thank you for your suggestion.We have incorporated your comments into the revised version.

Line408-411“[The pollen viability of M. savatieri exceeded 85% in both indoor and outdoor populations, as determined by MTT staining. This high viability ensures a sufficient supply of functional pollen grains, even under outdoor environmental stressors such as heavy rainfall.]"

Comments 7: -In point 3.3 Staining methods for the detection of suitable viability of pollen from M. savatieri, the description of staining methods for assessing pollen viability /lines 364-365, 374-375, 393-394, 399-401/ should be moved to “Materials and Methods”, section 4.4. Determination of pollen viability. Also, examples of the application of these staining methods to other species are unnecessary.

Response 7:  Thank you for your constructive feedback on our manuscript. We appreciate your suggestions regarding the structure and content of the section in question.We agree that this section can be shortened to focus on the most relevant findings.

Line494-504 Conclusions

“[M. savatieri is classified as a cross-pollinated post-flower species with monoecious flowers. Indoor cultivation is recommended for this plant, as environmental factors such as rainfall during flowering significantly reduce fruiting rates and challenge centralized harvesting in outdoor environments due to sequential fruit ripening and water-induced capsule dehiscence.

For pollen viability assessment, the MTT staining method was found to be more suitable than I2-KI or TTC staining. As a frequently heterogamous pollinated plant, systematic cross-breeding strategies should be implemented. However, to enhance breeding outcomes, parental plants must undergo rigorous self-crossing selection to eliminate inferior individuals and retain superior pure lines as hybrid parents.]"

Reviewer 3 Report

Comments and Suggestions for Authors

The authors aimed to provide a complete study of the flowering dynamic and pollination habits under artificial conditions of Monechasma savatieri, a popular medicinal plant in China. The morphological description and flowering dynamic sections are convincing, even if the images lack scale. The authors also provide a nice comparison of several staining methods to assess the pollen viability, However, it is a pity they never described any of these staining methods nor provide any references for them. The section dedicated to assessing the pollination biology of Monechasma savatieri is worth it. The material and methods are poor, I guess several necessary steps are missing (according to the manuscript, not any pollination nor crosses were done), and the authors do not master biological vocabulary, concepts and techniques linked to plant reproduction and pollination or at least the writing is very confusing, mixing words (they mention "self-intersection affinities", "fusionless reproduction", "success rate of heterosis", "natural heterosis" (without any link to hybrid production). Even if the authors want to study pollination biology, no observations were done on the seed set or seed quality (pollination rate, (viable seeds/ number of ovules), seed development, etc.…). They only recorded a fruit set.

I see two options for that paper: 1) Providing a shorter manuscript, keeping only the morphological description, the flowering dynamic section and the pollen viability analyses (if the related material and methods section is greatly improved).

2) Every section of the manuscript could be kept, but in addition to the improvements suggested in 1), all the sections about pollination biology have to be totally rethought to be coherent with the material and methods. Great care must be taken in the choice of words and the meaning of the biological concepts discussed.

Please find below some additional comments. I hope they will be received constructively and allow the improvement of the current manuscript.

Figure 1: Please provide a more precise legend, with a description of the different sections (a, b, c) of the figure and an explanation of the colour chart. Scale bars would also be helpful.

Figures 1, 2, 3, 4, 5, 6, 7, 8, 10, 11: Providing scale bars on the images would be nice.

Figure 12: It would be nice to have a more stand-alone legend providing the meanings of staining abbreviations. Moreover, the graph presents results at od, 3d, 14d, and 60d, not only at 60d, as presented in the legend.

L. 21 What is OCI? You have to explain abbreviations.

L 34 to 47 I am not sure listing all of the potential medicinal uses of the plant is fully useful here. However, providing a few key numbers, such as the global quantities produced per year or the value of the annual production of this plant, could help understand the economic value of this plant and, thus, the relevance of the present study.

L. 70 to 113, please add at least some references to Fig 1.

L. 99 to 113. Diagrams of the different floral organs (or photos of floral dissections) would be interesting to include to illustrate this part.

L. 194, 196, no space in "`s"

L. 224 what is the "self-intersecting" ? Would it be self-pollination?

L 225-226. No pollination, no seeds, so no partenogamy

L. 227 What is "fusionless reproduction"? I do not get it at all.

L. 228-229. your emasculation experiment demonstrates that M. savatieri can not produce parthenogamic seeds. I do not see the link with your post-flowering classification.

L. 229-230 As you never mention any hand pollination, if you only emasculated and bagged the plant, you are not doing cross-pollination; you can only see parthenogamy and parthenoicarpy. You never mention any hand pollination, pollen deposition on the stigmas, or anything else in the Material and Methods section.

If you want to study self-incompatibility and pollination deficit, you have to do hand-self and cross-pollination.

L 234 "the success of heterosis" NO you are not using the correct word: Maybe "pollination"? heterogamy? autogamy?

L. 237 What is "natural heterosis"??? Please check what is heterosis

Table 1, "setting rate", it would be more clear using "fruit set" or "fruiting rate" as in L. 229

L. 250 to 268 What are the abbreviations TTC, MTT, and FDA? Please explain them at the first occurrence.

L. 290 "heterosis" ? Maybe heterogamy?

L. 320 What is OCI? If you want to use an abbreviation, you have to explain it at the first occurrence.

L. 325-327. Please rephrase. The plant can not use artificial pollination to produce advantageous agronomic traits, but breeders can.

L. 366 to 376, 418, : All latin names have to be italics.

L. 402: Please remove the extra space before the ref [40]

L 444 to 456.

You wanted to test self-compatibility; what should be done by emasculating the plants and doing hand pollination with pollen from the same plant or from a different individual? The OCI values you use seem to assess the flower morphology adaptation to spontaneous self-pollination only. Not self-compatibility.

A1, you tested self-compatibility and spontaneous self-pollination (in the greenhouse)

A2, you want to test spontaneous pollination, but in a field environment. So, how did you protect the plant from cross-wind or insect pollination? There, I think you are not testing the spontaneous pollination.

B, you tested parthenogamy. But was it in a greenhouse or in the field? If it was in the field, how did you protect the plants against potential wind pollination?

C1 is the same as A1.

C2 You measured the current pollination in the field. By comparing C1 to C2, you can assess the importance of spontaneous pollination and the need for external pollen vectors (wind, insects, etc.…).

D you are testing spontaneous + wind pollination. Comparing D to C2, you can assess the need for insect pollination. 

What type of bags are you using? What mess size? What effect on wind flow? 

It is very surprising that you never did any hand pollination with pollen from the same plant and with pollen from another individual to test sel-compatibility and pollination deficit.

L 466: "the reproductive capability without fusion"? I have no idea what you are trying to speak about. Is it the capacity of spontaneous pollination without a pollen vector?

L 467-468, it is unclear which A and C you are using. A1? A2? C1? C2?

Moreover, are you really studying heterosis? Heterosis may be defined as the superiority of an F hybrid over both its parents in terms of yield or some other character. So, you probably want to study the ratio of self-pollination (autogamy) or cross-pollination (heterogamy). Or a deficit of pollination? But so you would have need some hand pollination.

L 446: a reference is missing for the OCI method

L 470 to 474. Please explain the different staining methods you used and provide references for them. 

It is a shame you did not do any seed observation to asses the seed set; it would have better reflected the quality of the pollination than the fruit set (except if there is a single seed per fruit?

Comments on the Quality of English Language

It is still unclear to me if section 2.2 is that confusing because of a misunderstanding of biological concepts or a significant mistranslation… (but even with mistranslation, major technical issues remain).

Author Response

Comments 1: Figure 1: Please provide a more precise legend, with a description of the different sections (a, b, c) of the figure and an explanation of the colour chart. Scale bars would also be helpful.

Figures 1, 2, 3, 4, 5, 6, 7, 8, 10, 11: Providing scale bars on the images would be nice.

Figure 12: It would be nice to have a more stand-alone legend providing the meanings of staining abbreviations. Moreover, the graph presents results at od, 3d, 14d, and 60d, not only at 60d, as presented in the legend.

Response 1: Thank you for your detailed feedback on our manuscript and the figures. We appreciate your suggestions.

Regarding Figure 1, we will revise the legend to provide a more precise description of each section (a, b, c). Additionally, scale bars will be added to enhance the figure's interpretability.

For Figures 1, 2, 3, 4, 5, 6, 7, 8, 10, and 11, we acknowledge your suggestion and will add scale bars to all images to ensure consistency and aid in the accurate interpretation of the data.

“[Figures 1, 2, 3, 4, 5, 6, 7, 8, 10, 1112]”

Comments 2:  It would be nice to have a more stand-alone legend providing the meanings of staining abbreviations. Moreover, the graph presents results at od, 3d, 14d, and 60d, not only at 60d, as presented in the legend.

Response 2: Concerning Figure 12, we will update the legend to make it more stand-alone by clearly explaining the meanings of the staining abbreviations. Furthermore, we will ensure that the legend accurately reflects the data presented at all time points (od, 3d, 14d, and 60d), not just at 60d, to provide a comprehensive overview of the results.

“[Figures 12]”

Comments 3: What is OCI? You have to explain abbreviations.

Response 3:  Thank you for pointing this out. We agree with this comment. Therefore, we  explicitly define outcrossing index (OCI)  in line 21. And at Materials and Method stage, relevant references were added.

line 21“[The plant, with an outcrossing index (OCI) of 3, ]”

Comments 4: L 34 to 47 I am not sure listing all of the potential medicinal uses of the plant is fully useful here

Response 4: Thank you for your suggestion. We have revised the manuscript to address this point. To streamline the introduction, we will shorten the section on pharmaceutical uses and active compounds, ensuring it remains concise and relevant to the study's objectives.

Line34-38“[M. savatieri, a traditional Chinese medicinal herb, is used holistically to treat conditions such as respiratory infections, inflammatory disorders, and gynecological ailments [1]. Its key bioactive components—flavonoids, alkaloids, saponins, and polysaccharides [2,5]—contribute to broad pharmacological activities, including antimicrobial, antioxidant, and anti-inflammatory effects [4,6–15]. Recent studies emphasize its anticancer potential [16].]”

Comments 5: L. 70 to 113, please add at least some references to Fig 1.

L. 99 to 113. Diagrams of the different floral organs (or photos of floral dissections) would be interesting to include to illustrate this part.

Response 5: Thank you for your constructive comments on our manuscript. We are grateful for your suggestions.

Regarding the section on lines 70 through 113, we will refer to fig. 1 to give the reader a clearer idea of the connection between the described floral features and the pictures. This will help to illustrate the main points discussed in this section.

In addition, we agree that the inclusion of diagrams or photographs of the different floral organs, as well as an anatomical drawing of the flower, will provide a more complete illustration of floral morphology and development.

“[2.1.1;Figure 1.]”

Comments 6: L. 194, 196, no space in "`s"

Response 6: Thank you for pointing this out.

Line223-227“[Within two days, the stigma adjusts its position to match the flower’s depth (Fig. 8). ]”

Comments 7: L. 224 what is the "self-intersecting" ? Would it be self-pollination?

Response 7: Thanks for pointing out the problem, we changed it to “ self-compatibility”.

Line222“[2.2 The self-compatibility of M. savatieri]”

Comments 8: L .225-226. No pollination, no seeds, so no partenogamy

L. 227 What is "fusionless reproduction"? I do not get it at all.

L. 228-229. your emasculation experiment demonstrates that M. savatieri can not produce parthenogamic seeds. I do not see the link with your post-flowering classification.

Response 8:

Thank you for your valuable comments and suggestions regarding our manuscript. We have optimized the expression of the experimental conclusions.

Line194-195“[The study demonstrated that M. savatieri, when subjected to artificial emasculation and bagging, failed to produce fruit, resulting in a D/A ratio of 0. This result indicates that M. savatieri is unable to produce seeds through apomixis and relies on pollination for reproduction.]”

Comments 9: L. 229-230 As you never mention any hand pollination, if you only emasculated and bagged the plant, you are not doing cross-pollination; you can only see parthenogamy and parthenoicarpy. You never mention any hand pollination, pollen deposition on the stigmas, or anything else in the Material and Methods section.If you want to study self-incompatibility and pollination deficit, you have to do hand-self and cross-pollination.

Response 9:

Thank you for your insightful comments. We acknowledge the oversight in not mentioning artificial pollination in the Methods section. To address this, we will revise the manuscript to include a description of the artificial pollination experiments. This will clarify our approach and strengthen the study's methodology.

Line231-233“[Additionally, artificial pollination experiments revealed that artificial self-pollination resulted in a fruiting rate of 85.29%, while artificial cross-pollination achieved a higher fruiting rate of 90.00%.]”

Comments 10: L 234 "the success of heterosis" NO you are not using the correct word: Maybe "pollination"? heterogamy? autogamy?

L.237 What is "natural heterosis"??? Please check what is heterosis

L. 290 "heterosis" ? Maybe heterogamy?

Response 10: Thank you for pointing out the inaccuracy in our terminology. Upon reviewing the manuscript, we realize that the term "heterosis" was incorrectly used. We have revised the text to replace "heterosis" with "outcrossing" to accurately reflect the context of our study. Additionally, we acknowledge your comment regarding "natural heterosis" and have removed this term entirely from the manuscript to avoid confusion.

Line297-299[Delayed self-fertilization is a widely recognized reproductive adaptation that ensures seed production in the absence of pollinators while prioritizing outcrossing when pollinators are abundant.]

Line321-322“[For example, increasing the distance between Setaria plantings reduces the likelihood of outcrossing,]”

Comments 11: Table 1, "setting rate", it would be more clear using "fruit set" or "fruiting rate" as in L. 229

Response 11: Thank you for your comments. We have revised the terminology in Table 1 from "setting rate" to "fruiting rate" to maintain consistency with the terminology used in the text (as seen in line 229). This change enhances the clarity and readability of the table.

Comments 12: L. 250 to 268 What are the abbreviations TTC, MTT, and FDA? Please explain them at the first occurrence.

Response 12: Thank you for your constructive feedback.

Regarding the abbreviations TTC, MTT, and FDA mentioned in lines 250 to 268, we acknowledge your suggestion to provide their full explanations upon their first occurrence. To address this, we have revised the manuscript to include the full names of these abbreviations when they are first introduced. Specifically:

TTC: 2, 3, 5-Triphenyltetramlium chloride

MTT: 3-(4,5-Dimethylthiazol-2-yl)-2,5-diphenyltetrazolium bromide

FDA: Fluorescein Diacetate

Comments 13: L. 320 What is OCI? If you want to use an abbreviation, you have to explain it at the first occurrence.

L 446: a reference is missing for the OCI method

Response 13: Thank you for pointing this out. We agree with this comment. Therefore, we  explicitly define outcrossing index (OCI)  in line 21. And at Materials and Method stage, relevant references were added.

line 21“[The plant, with an outcrossing index (OCI) of 3, ]”

Comments 14: L. 325-327. Please rephrase. The plant can not use artificial pollination to produce advantageous agronomic traits, but breeders can.

Response 14:

Thank you for your feedback. We agree with your suggestion and have rephrased the sentences to clarify the distinction between artificial pollination and breeding objectives.

Line335-338“[Among flowering plants, hybridization is a key strategy employed by breeders to develop progeny with advantageous agronomic traits or enhanced resistance to abiotic and biotic stresses. This process is typically facilitated through artificial pollination techniques.]”

Comments 15: You wanted to test self-compatibility; what should be done by emasculating the plants and doing hand pollination with pollen from the same plant or from a different individual? The OCI values you use seem to assess the flower morphology adaptation to spontaneous self-pollination only. Not self-compatibility.gure 1:

Response 15:

Thank you for your insightful comments.

In this section, we aimed to explore the reproductive biology of M. savatieri by examining its floral morphology and its potential implications for pollination strategies. The experiment was designed to assess whether the plant exhibits traits that might facilitate or hinder self-pollination. By emasculating plants and performing artificial pollination with pollen from the same plant, we sought to test the plant's self-compatibility and its ability to produce viable seeds through self-pollination.

The OCI values we used are intended to reflect the plant's morphological adaptations to spontaneous self-pollination. However, we acknowledge your point that these values may not directly assess self-compatibility. To address this, we have conducted additional artificial pollination experiments, including both self-pollination and cross-pollination, to validate the predictions derived from the floral morphological analysis.

In addition, we provide a clearer description of the pollination experiment.

Line249-254,462-467“[The A1 group was natural pollination in a greenhouse environment (CK1), The A2 group was natural pollination in a field environment (CK2), The B group was directly bagged, The C1 group emasculation without bagging in the greenhouse, The C2 group emasculation without bagging in the field, The D group emasculation and bagged, The E group of artificial geitonogamy(self-pollination within the same flower), The F group of artificial xenogamy.]”

Comments 16: What type of bags are you using? What mess size? What effect on wind flow? 

Response 16:

Bagging is done in 6 cm × 10 cm sulphate paper bags, which are not affected by wind after bagging.

Line468-469“[Bagging is done in 6 cm × 10 cm sulphate paper bags, with each flower individually segregated in a bag.]”

Comments 17: It is very surprising that you never did any hand pollination with pollen from the same plant and with pollen from another individual to test sel-compatibility and pollination deficit.

Response 17:

Thank you for your insightful suggestion. In response, we have added treatments involving artificial self-pollination and cross-pollination to the manuscript.

Table 1“[The E group of artificial geitonogamy(self-pollination within the same flower), The F group of artificial xenogamy.]”

Comments 18: L 466: "the reproductive capability without fusion"? I have no idea what you are trying to speak about. Is it the capacity of spontaneous pollination without a pollen vector?

Response 18:

Thank you for highlighting this ambiguity. We agree that the original phrasing was unclear and have revised the sentence to explicitly state the focus on apomixis (asexual seed production without fertilization).

This adjustment clarifies that the study specifically tested for apomictic reproduction (via the D/A ratio method), distinguishing it from spontaneous self-pollination or vector-dependent processes. The term "D/A ratio" refers to the proportion of seeds produced in emasculated, bagged flowers (no pollination allowed, D) relative to open-pollinated controls (A). A D/A ratio of 0 confirms the absence of apomixis.

Line469-471“[The study primarily examined fruiting rate. The ability of M. savatieri to reproduce through apomixis was evaluated by calculating the ratio of D/A. ]”

Comments 19: L 467-468, it is unclear which A and C you are using. A1? A2? C1? C2?

Moreover, are you really studying heterosis? Heterosis may be defined as the superiority of an F hybrid over both its parents in terms of yield or some other character. So, you probably want to study the ratio of self-pollination (autogamy) or cross-pollination (heterogamy). Or a deficit of pollination? But so you would have need some hand pollination.

Response 19:

Thank you for your detailed feedback and for pointing out the ambiguity in our use of variables A and C. Upon reviewing our manuscript, we realize that the notation was insufficiently clear. To address this, we have revised the sections to explicitly define A and E as follows

Line249-254,462-467“[The A1 group was natural pollination in a greenhouse environment (CK1), The A2 group was natural pollination in a field environment (CK2), The B group was directly bagged, The C1 group emasculation without bagging in the greenhouse, The C2 group emasculation without bagging in the field, The D group emasculation and bagged, The E group of artificial geitonogamy(self-pollination within the same flower), The F group of artificial xenogamy.]”

Comments 20: L 470 to 474. Please explain the different staining methods you used and provide references for them. 

Response 20:

Thank you for your feedback and for highlighting the need for a clearer explanation of the staining methods used in our study. In response to your comment, we have revised the manuscript to include a detailed description of the different staining methods used to assess pollen viability. Specifically, we have provided the following explanations and references:

Line480-493“[The I2-KI staining method is based on the accumulation of starch in pollen, with blue-stained pollen considered viable [24].

The TTC (2, 3, 5-Triphenyltetramlium chloride) method, which induces a red coloration through a reaction with succinate dehydrogenase in viable cells, is widely used for pollen viability assessment [26].

Aceto carmine staining targets the nuclei of pollen cells and stains viable pollen red,  has demonstrated clear and distinguishable results in species such as peanut [27].

MTT (3-(4,5)-dimethylthiahiazo (-z-y1)-3,5-di-phenytetrazoliumromide) is a yellow-colored dye. MTT colorimetric assay can detecting cell survival and growth. The principle of the assay is that the enzyme succinate dehydrogenase in the mitochondria of living cells reduces exogenous MTT to the water-insoluble blue-violet crystalline formazan and deposits it in the cells, whereas dead cells do not have this function [28].

The FDA (fluorescein diacetate) reacting with pollen grain esterases to produce fluorescein, thereby enabling staining and viability assessment [26]. ]”

Comments 21: It is a shame you did not do any seed observation to asses the seed set; it would have better reflected the quality of the pollination than the fruit set (except if there is a single seed per fruit?

Response 21:

We concur with the reviewer that seed set observation could provide additional insights into pollination quality assessment. However, due to limitations such as insufficient sample preservation for seed counting, time constraints during the field observation period, and the current unavailability of specialized equipment for seed viability testing, it is difficult to supplement these data in the present study. In future studies, we will prioritize the use of multidimensional evaluation metrics, including seeds, to enhance pollination biology research.

4. Response to Comments on the Quality of English Language

Point 1:It is still unclear to me if section 2.2 is that confusing because of a misunderstanding of biological concepts or a significant mistranslation… (but even with mistranslation, major technical issues remain).

Response 1:  We appreciate the reviewer’s attention to clarity in Section 2.2. We acknowledge that potential ambiguities in phrasing may have arisen during translation, and we have carefully re-examined the original text to ensure alignment with the intended biological concepts. We have also revised the section to enhance terminological precision and methodological transparency. Should further clarification be needed, we are happy to provide additional details or schematic illustrations of the workflow. Moving forward, we will prioritize bilingual peer-review for critical technical sections to minimize linguistic or conceptual misinterpretations.

Round 2

Reviewer 3 Report

Comments and Suggestions for Authors

The authors considered all the comments and provided constructive elements to improve the manuscript. The experimental setting is much more convincing and easy to follow, even if the ovule number is still not mentioned. It would have been helpful to provide an idea of the seed number expected by the flower, which is very useful when you want to speak about breeding. But that would request another fieldwork season for data acquisition. The authors added scales to all the images, but they are small and not that easy to read. It would be nice to make the scales more easy to read.

I just have noticed some typos in the manuscript:

Fig 3. M. savatieri should be in italics.

L. 303 Hibiscus trionum should be in italics.

L. 308 Viola pubescens should be in italics.

L. 320 "an" instead of "a"

L. 321 "an" instead of "a"

L. 391 a dot is missing after "species"

L. 419-420 Gardenie jasminoides should be in italics.

L. 77, 79, 82, 86, 448, a space is missing before the reference to figure or paper

Author Response

We sincerely appreciate the reviewer’s constructive feedback and acknowledge the importance of the points raised. Below, we address the concerns raised in the comments:

Comments 1: It would have been helpful to provide an idea of the seed number expected by the flower, which is very useful when you want to speak about breeding. But that would request another fieldwork season for data acquisition.

Response 1: We fully agree that ovule number and expected seed production per flower are critical parameters for breeding studies. However, as the reviewer noted, collecting these data would require an additional fieldwork season to track post-pollination seed development, which was beyond the scope of the current study.

 We added the number of seeds cultivated indoors and will focus on the number of ovules per flower and expected seed production in subsequent studies.

Line136“[Each capsule contained 72.57 ±16.54 seeds]”

Comments 2: The authors added scales to all the images, but they are small and not that easy to read. It would be nice to make the scales more easy to read.

Response 2: Thank you for highlighting the readability of scale bars in the figures. We have revised all images to enlarge the scale bars and adjusted font sizes for improved visibility.

“[figure1,2,3,4,5,6,8,10,11]”

Comments 3: I just have noticed some typos in the manuscript:

Fig 3. M. savatieri should be in italics.

L. 303 Hibiscus trionum should be in italics.

L. 308 Viola pubescens should be in italics.

L. 320 "an" instead of "a"

L. 321 "an" instead of "a"

L. 391 a dot is missing after "species"

L. 419-420 Gardenie jasminoides should be in italics.

L. 77, 79, 82, 86, 448, a space is missing before the reference to figure or paper

Response 3: We appreciate your careful review of the manuscript. All typos and content errors you highlighted have been corrected in the revised version.

“[Fig 3. M. savatieri has been changed to italics.

L. 305 Hibiscus trionum has been changed to italics.

L. 310 Viola pubescens has been changed to italics.

L. 321 "an" instead of "a"

L. 322 "an" instead of "a"

L. 393 a dot is added after "species"

L. 421-422 Gardenie jasminoides has been changed to italics.

L. 77, 79, 82, 86, 450, a space is added before the reference to figure or paper]”
